# Decoupled Diffusion Models for Efficient Spatio-Temporal Graph Forecasting

## Abstract

Graph-based diffusion models suffer from a critical computational bottleneck, limiting their use in practical applications such as spatio-temporal graph forecasting. We argue that this inefficiency stems from the fusion of information propagation and feature transformation within standard GNNs. In this paper, we introduce a design principle that decouples these two operations, enabling a highly efficient and linear architecture. Instantiating this principle, Decoupled Spatio-Temporal Diffusion Model (DSTD) leverages the principle alongside a dynamic multi-scale aggregation mechanism to achieve remarkable performance. On widely-used spatio-temporal graph forecasting benchmarks, DSTD not only outperforms existing probabilistic methods but also surpasses top-performing deterministic models, while demonstrating a significant reduction in inference time. Our results validate that decoupling is a powerful and effective strategy for building scalable and high-performing generative models for graph-structured data.

## 1 Introduction

Deep generative models, particularly diffusion models (Sohl-Dickstein et al., 2015; Song & Ermon, 2019; Ho et al., 2020), have achieved unprecedented success in modeling continuous, high-dimensional data such as images (Ramesh et al., 2021; Rombach et al., 2022b) and audio (Liu et al., 2023; Ju et al., 2024). The next frontier for generative modeling is effectively capturing the complexities of relational data, specifically graphs (Hudovernik et al., 2025). However, generative modeling for graphs, especially conditional generation where the process is guided by complex contexts such as historical time series, presents a fundamental challenge. This challenge is particularly pronounced when modeling spatio-temporal graphs, where dynamic temporal patterns are intricately entangled with complex spatial dependencies (Yang et al., 2024). This represents one of the most complex and demanding scenarios for generative models to tackle.

Applying diffusion models to graphs typically involves employing a Graph Neural Network (GNN) (Kipf & Welling, 2017; Hamilton et al., 2017; Veličković et al., 2018) as the core of the denoising network (Kong et al., 2023; Yang et al., 2023). However, the generative process of a diffusion model is inherently iterative, consisting of hundreds of denoising steps to reverse the noise process (Ho et al., 2020). Repeatedly invoking a computationally expensive GNN within this loop creates a severe computational bottleneck, making the training and inference of these models remarkably slow. This bottleneck is especially crippling for tasks such as spatio-temporal graph forecasting, which demand predictions over long horizons and for a large number of nodes (Wen et al., 2023). The necessity of running GNN operations over numerous nodes hundreds of times makes existing approaches impractical and severely limits their scalability and real-world applicability.

To fundamentally address this computational bottleneck, we propose a new architectural design principle for GNNs within diffusion models: the decoupling of information propagation and feature transformation. First, a parameter-free graph convolution operation handles only the structural information propagation, efficiently gathering information from neighbors across various scales (hops). Second, lightweight feature transformation modules are responsible for the actual learnable transformations on this rich, propagated information. This separation of duties dramatically reduces computational complexity while effectively preserving the expressive power of the GNN, making it an ideal structure for the iterative generation process of diffusion models.

We instantiate this design principle in a new framework, Decoupled Spatio-Temporal Diffusion Model (DSTD), which is tailored for spatio-temporal data. The condition and denoising networks in DSTD are both architected to efficiently handle spatial propagation and temporal feature learning in a decoupled manner. Notably, DSTD features a dynamic multi-scale aggregation mechanism. By applying a weighted sum to the output of the intermediate blocks, the model learns to dynamically select the optimal receptive field for the prediction task. Furthermore, we introduce a learnable position-aware embedding to efficiently inject node-specific inductive biases, enhancing the expressive power of the model with minimal overhead. To demonstrate the power and efficiency of our design principle, we evaluate DSTD on one of the most demanding benchmarks in generative modeling: probabilistic spatio-temporal graph forecasting. Through extensive experiments, DSTD achieves state-of-the-art or highly competitive performance in both probabilistic and deterministic forecasting, while exhibiting significantly faster inference than existing complex diffusion-based methods. Our contributions are summarized as follows:

- We introduce a new, highly efficient architectural design principle for graph-based diffusion models based on the decoupling of propagation and transformation.

- We propose the DSTD framework, which realizes this principle with a dynamic multi-scale aggregation mechanism.

- We confirm the superiority of our approach through extensive experiments on challenging real-world spatio-temporal graph benchmarks.

## 2 PRELIMINARIES

### 2.1 PROBLEM FORMULATION

Let $\mathcal{G} = \{\mathcal{V}, \mathcal{E}\}$ represent a graph, where $\mathcal{V}$ and $\mathcal{E}$ are sets of nodes and edges, respectively. $N = |\mathcal{V}|$ is the number of nodes in the graph. The graph includes node features $X \in \mathbb{R}^{N \times T \times d_x}$, where $T$ is the time window and $d_x$ is the number of variables. Edges can also be represented as an adjacency matrix $A \in \mathbb{R}^{N \times N}$, where $A_{ij} = 1$ when there is an edge between node $i$ and $j$, and otherwise 0. Spatio-temporal graph forecasting focuses on learning a function $\mathcal{F}$ that predicts future variables $X^p$ given a graph $\mathcal{G}$ and a history $X^h$. The prediction function $\mathcal{F}$ is formulated as follows:

$$\mathcal{F} : (X^h, \mathcal{G}) \to X^p, \quad X^h \in \mathbb{R}^{N \times T_h \times d_x}, X^p \in \mathbb{R}^{N \times T_p \times d_x}, \tag{1}$$

where $T_h$ and $T_p$ are the history and prediction lengths, respectively.

### 2.2 DENOISING DIFFUSION PROBABILISTIC MODELS

Denoising Diffusion Probabilistic Models (DDPMs) Ho et al. (2020) learn the distribution $p_\theta(X_0)$ to approximate the data distribution $q(X_0)$ and generate new samples. These models consist of forward and reverse processes. The forward process gradually adds noise to the data. The reverse process iteratively removes the noise to recover the original data. Owing to this denoising process, DDPMs can generate high-resolution samples with fine-grained details. The forward process is a Markov chain that incrementally adds Gaussian noise to the data over $K$ steps as follows:

$$q(X_{1:K}|X_0) = \prod_{k=1}^{K} q(X_k|X_{k-1}), \tag{2}$$

where each transition in the chain follows a Gaussian distribution:

$$q(X_k|X_{k-1}) = \mathcal{N}(X_k; \sqrt{1 - \beta_k}X_{k-1}, \beta_k I). \tag{3}$$

Note that $\beta_k \in (0, 1)$ is a variance that controls the noise level at each step. $X_k$ can be directly calculated from the original data $X_0$ as follows:

$$X_k = \sqrt{\alpha_k}X_0 + \sqrt{1 - \alpha_k}\epsilon, \quad \epsilon \sim \mathcal{N}(0, I), \tag{4}$$

where $\alpha_k = \prod_{i=1}^{k}(1 - \beta_i)$ represents the cumulative noise scale up to step $k$. After a sufficient number of steps, the data distribution $q(X_K)$ becomes $\mathcal{N}(0, I)$, which is independent of the data.

The reverse process aims to recover the original data $X_0$ by iteratively removing the noise from $X_K$. With Gaussian noise $X_K \sim \mathcal{N}(0, I)$, the reverse process follows a Markov chain:

$$p_\theta(X_{0:K}) = p(X_K) \prod_{k=1}^{K} p_\theta(X_{k-1}|X_k), \tag{5}$$

$$p_\theta(X_{k-1}|X_k) = \mathcal{N}(X_{k-1}; \mu_\theta(X_k, k), \sigma_\theta(X_k, k)), \tag{6}$$

where $\mu_\theta(X_k, k)$ and $\sigma_\theta(X_k, k)$ are the parameterized mean and variance, respectively. Starting from arbitrary Gaussian noise, DDPMs generate new samples through a repeated reverse process.

The training objective is derived from the variational lower bound on the negative log-likelihood, which is written as follows:

$$\mathcal{L}_{VLB} = \mathbb{E}_{q(X_{0:K})} \left[ -\log p(X_K) - \sum_{k=1}^{K} \log \frac{p_\theta(X_{k-1}|X_k)}{q(X_k|X_{k-1})} \right]. \tag{7}$$

The objective is reformulated as a noise prediction task to simplify the training. Instead of directly modeling $p_\theta(X_{k-1}|X_k)$, the denoising network $\epsilon_\theta(X_k, k)$ is introduced to predict the added noise $\epsilon$ in the forward process. The final training objective is as follows:

$$\mathcal{L} = \mathbb{E}_{X_0, \epsilon, k} \left[ ||\epsilon - \epsilon_\theta(X_k, k)||^2 \right]. \tag{8}$$

This enables stable training by leveraging the mean squared error loss.

### 2.3 ACCELERATED INFERENCE

The iterative inference process has a significant computational overhead. Song et al. (2021) proposed an accelerated inference that reduces the number of required steps. They sample $M$ steps out of $K$ diffusion steps and skip the remaining intermediate steps. With the set $\{\tau_1, \ldots, \tau_M\}$, the transitions are modified as follows in the accelerated inference:

$$X_{\tau_{m-1}} = \sqrt{\alpha_{\tau_{m-1}}} \hat{X}_0 + \sqrt{1 - \alpha_{\tau_{m-1}} - \sigma_{\tau_m}^2} \epsilon_\theta(X_{\tau_m}, \tau_m) + \sigma_{\tau_m} \epsilon_{\tau_m}, \tag{9}$$

where $\epsilon_{\tau_m} \sim \mathcal{N}(0, I)$ and $\hat{X}_0$ is the estimated data, computed as follows:

$$\hat{X}_0 = \frac{X_{\tau_m} - \sqrt{1 - \alpha_{\tau_m}} \epsilon_\theta(X_{\tau_m}, \tau_m)}{\sqrt{\alpha_{\tau_m}}}. \tag{10}$$

In this study, we set $\sigma_{\tau_m} = \sqrt{(1 - \alpha_{\tau_{m-1}})/(1 - \alpha_{\tau_m})} \sqrt{1 - \alpha_{\tau_m}/\alpha_{\tau_{m-1}}}$ to match DDPMs, where $\sigma_{\tau_m}$ controls the level of inference stochasticity.

## 3 METHOD

In this section, we introduce Decoupled Spatio-Temporal Diffusion Model (DSTD), a new framework designed for efficient and accurate probabilistic spatio-temporal graph forecasting. The architecture of DSTD is built upon two main components: a condition network and a denoising network. Figure 1 illustrates the overall framework of DSTD. The condition network first encodes the historical observations $X^h$ into a concise condition representation $C_\phi$. This representation guides the denoising network, which iteratively refines noisy data to generate the final forecast. Both networks share a learnable position-aware embedding $P_\psi$. This embedding provides unique, node-specific features that capture static spatial characteristics, ensuring that the model can differentiate between nodes with distinct structural roles or temporal patterns.

### 3.1 DECOUPLED PROPAGATION AND TRANSFORMATION

The core of efficiency and expressive power of DSTD lies in our proposed design principle: the decoupling of information propagation and feature transformation. Unlike standard GCN layers that fuse these two operations, we separate them into distinct, specialized layers. This design is particularly advantageous for the iterative nature of diffusion models.

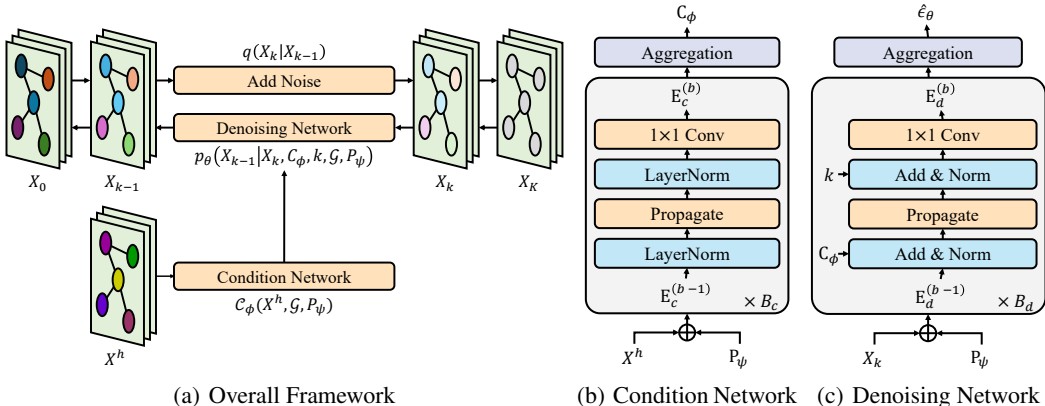

(a) Overall Framework    (b) Condition Network    (c) Denoising Network

Figure 1: The architecture of Decoupled Spatio-Temporal Diffusion Model (DSTD). (a) The overall framework, where the condition network encodes historical graph data to guide the denoising network. (b) The detailed structure of the decoupled condition network. (c) The structure of the conditional decoupled denoising network, illustrating how guidance is injected.

For information propagation layer, we employ a parameter-free graph convolution. This layer aggregates structural information from neighboring nodes without performing any feature transformation. This is achieved by multiplying the node features by the normalized adjacency matrix as follows:

$$M^{(b)} = \tilde{A} \cdot \text{LayerNorm}(E^{(b-1)}), \tag{11}$$

where $\tilde{A} = D^{-\frac{1}{2}} \hat{A} D^{-\frac{1}{2}}$ is a normalized adjacency matrix with self-loop $\hat{A} = A + I$, and $E^{(b-1)}$ is the intermediate output of $(b-1)$-th block. This parameter-less approach prevents the information distortion that can occur during weighted transformations and reduces computational cost.

In the transformation layer, the feature transformation is handled by lightweight modules, specifically a $1 \times 1$ convolution applied across the time dimension, after the propagation step:

$$E^{(b)} = \text{Conv}_{1 \times 1}(\text{LayerNorm}(M^{(b)})). \tag{12}$$

By decoupling these operations, we create a highly efficient architectural block that effectively captures spatial and temporal dependencies. A key aspect of this design is its inherent linearity. Through deliberately omitting non-linear activation functions, we preserve the smooth flow of information across layers, which is effective in deep graph models. This concise and linear design minimizes computational overhead, making it ideal for repeated execution within the diffusion process.

## 3.2 DYNAMIC MULTI-SCALE AGGREGATION

Stacking our decoupled blocks allows the model to capture information from varying neighborhood sizes. The output of the $b$-th block $E^{(b)}$ effectively contains information aggregated from a $b$-hop receptive field. Instead of relying solely on the output of the final block, DSTD leverages a dynamic multi-scale aggregation mechanism to combine information from all intermediate blocks. This is achieved using a learnable weighted sum of all block outputs, which is computed as follows:

$$Z = \text{MLP}\left(\sum_{b=0}^{B} \alpha^{(b)} E^{(b)}\right), \tag{13}$$

where MLP is a multi-layer perceptron and $B$ is the number of blocks. $\alpha^{(b)}$ is a learnable weight, derived from a softmax function, representing the importance score of the $b$-hop representation. This mechanism enables the model to dynamically select and fuse the most relevant spatial scales for the given task, granting it greater flexibility and adaptability compared to fixed-depth GNNs.

### 3.3 DECOUPLED SPATIO-TEMPORAL DIFFUSION MODEL

We detail how the general principles of our decoupled architecture and dynamic aggregation are instantiated within the two components of DSTD: the condition network and the denoising network.

**Condition Network** The condition network is responsible for encoding the historical observations $X^h$ into a compact and informative condition $C_\phi$. It begins by projecting the input history $X^h$ and the shared position-aware embedding $P_\psi$ into a common latent space. These projected representations are combined using element-wise addition to form the initial representation $E_c^{(0)}$:

$$E_c^{(0)} = \text{Proj}_c^{in}(X^h) + \text{Proj}_c^{pos}(P_\psi). \tag{14}$$

This combined representation is processed through a stack of $B_c$ decoupled blocks. We apply the dynamic multi-scale aggregation mechanism to the intermediate block outputs $(E_c^{(0)}, \dots, E_c^{(B_c)})$ to produce the final condition representation $C_\phi$. This representation encapsulates the essential spatio-temporal patterns from the history, which guides the generative process of the denoising network.

**Denoising Network** The goal of the denoising network is to predict the noise $\epsilon$ that is added to an input $X_0$ to create the noisy data $X_k$, conditioned on the guidance from the condition network. Similar to the condition network, it forms an initial representation $E_d^{(0)}$ by projecting the noisy input $X_k$ and the shared position-aware embedding $P_\psi$ into a common latent space as follows:

$$E_d^{(0)} = \text{Proj}_d^{in}(X_k) + \text{Proj}_d^{pos}(P_\psi). \tag{15}$$

This initial representation is processed through a series of $B_d$ decoupled blocks. Crucially, each block is conditioned on the $C_\phi$ from the condition network and the diffusion timestep $k$. The input representation $E_d^{(b-1)}$ of the $b$-th denoising block is enriched with the condition $C_\phi$ before being passed to the propagation layer. The spatial information propagation is formulated as follows:

$$M_d^{(b)} = \tilde{A} \cdot \text{LayerNorm}(E_d^{(b-1)} + \text{Proj}_d^c(C_\phi)). \tag{16}$$

The sinusoidal diffusion step embedding $\mathcal{S}(k)$ is added to the spatially-aggregated message. The transformation layer extracts relevant features from the combined information as follows:

$$E_d^{(b)} = \text{Conv}_{1\times1}(\text{LayerNorm}(M_d^{(b)} + \text{Proj}_d^k(\mathcal{S}(k)))). \tag{17}$$

The final predicted noise $\hat{\epsilon}_\theta$ is obtained by using the dynamic multi-scale aggregation mechanism to all intermediate block outputs $(E_d^{(0)}, \dots, E_d^{(B_d)})$. This conditional block structure effectively utilizes historical context and temporal information at each step of the denoising process.

### 3.4 OVERALL DIFFUSION PROCESS

We integrate the decoupled networks into a conditional diffusion process. The reverse process of DSTD is conditioned on the graph structure $\mathcal{G}$ and the condition $C_\phi$, which is formulated as follows:

$$p_\theta(X_{0:K}|C_\phi, \mathcal{G}) = p(X_K) \prod_{k=1}^{K} p_\theta(X_{k-1}|X_k, C_\phi, k, \mathcal{G}, P_\psi). \tag{18}$$

The training objective is formulated to minimize the mean squared error between the true noise $\epsilon$ and the predicted noise by the denoising network $\epsilon_\theta$ as follows:

$$\mathcal{L} = \mathbb{E}_{X_0,\epsilon,k}[||\epsilon - \epsilon_\theta(X_k, C_\phi, k, \mathcal{G}, P_\psi)||^2], \tag{19}$$

where $X_k$ is computed using Equation 4. This objective enables stable training of the entire DSTD framework. To achieve fast predictions, we adopt the accelerated inference described in Section 2.3. Accelerated conditional reverse transition with the set $\{\tau_1, \dots, \tau_M\}$ is represented as follows:

$$X_{\tau_{m-1}} = \sqrt{\alpha_{\tau_{m-1}}}\hat{X}_0 + \sqrt{1 - \alpha_{\tau_{m-1}} - \sigma_{\tau_m}^2}\hat{\epsilon}_\theta + \sigma_{\tau_m}\epsilon_{\tau_m}, \tag{20}$$

$$\hat{X}_0 = \frac{X_{\tau_m} - \sqrt{1 - \alpha_{\tau_m}}\hat{\epsilon}_\theta}{\sqrt{\alpha_{\tau_m}}}, \tag{21}$$

where $\hat{\epsilon}_\theta = \epsilon_\theta(X_{\tau_m}, C_\phi, \tau_m, \mathcal{G}, P_\psi)$ is the predicted noise of the denoising network. The complete procedures for training and accelerated inference are detailed in Algorithm 1 and 2, respectively.

**Algorithm 1** Training of DSTD

**Input:** Data distribution $q(X)$, graph $\mathcal{G}$
**Output:** Trained condition function $\mathcal{C}_\phi(\cdot)$, trained denoising function $\epsilon_\theta(\cdot)$, trained position-aware embedding $P_\psi$
 1: **repeat**
 2:     Sample $X_0 \sim q(X), k \sim \mathcal{U}(1, K), \epsilon \sim \mathcal{N}(0, I)$
 3:     Get $X^h$ from the dataset given $X_0$
 4:     Compute $C_\phi = \mathcal{C}_\phi(X^h, \mathcal{G}, P_\psi)$
 5:     Compute $X_k = \sqrt{\alpha_k}X_0 + \sqrt{1 - \alpha_k}\epsilon$
 6:     Take a gradient descent on Equation 19
 7: **until** converged

**Algorithm 2** Inference of DSTD

**Input:** History $X^h$, graph $\mathcal{G}$, condition function $\mathcal{C}_\phi(\cdot)$, denoising function $\epsilon_\theta(\cdot)$, position-aware embedding $P_\psi$
**Output:** Inference target $X^p$
 1: Sample $X_{\tau_M} \sim \mathcal{N}(0, I)$
 2: Compute condition $C_\phi = \mathcal{C}_\phi(X^h, \mathcal{G}, P_\psi)$
 3: **for** $m = M$ to 1 **do**
 4:     Sample $\epsilon_{\tau_m} \sim \mathcal{N}(0, I)$
 5:     Compute $\hat{X}_0$ using Equation 21
 6:     Compute $X_{\tau_{m-1}}$ using Equation 20
 7: **end for**
 8: **return** $X_0$

## 4 EXPERIMENT

### 4.1 EXPERIMENTAL SETUP

**Datasets**  We evaluate DSTD on two widely-adopted real-world traffic forecasting benchmarks: METR-LA and PEMS-BAY Li et al. (2018). The METR-LA dataset contains traffic speed data from 207 sensors on Los Angeles County highways. The PEMS-BAY dataset is collected from 325 sensors in the San Francisco Bay Area. For both datasets, the road network is used to construct the graph structure. Table 1 presents the detailed statistics of the datasets.

Table 1: Statistics of the datasets. # denotes the number of.

| Dataset | # Nodes | # Edges | Time Range | Time Interval | # Frames |
|---------|---------|---------|------------|---------------|----------|
| METR-LA | 207 | 1,515 | 03/01/2012 - 06/27/2012 | 5 min | 34,272 |
| PEMS-BAY | 325 | 2,369 | 01/01/2017 - 06/30/2017 | 5 min | 52,116 |

**Baselines**  To comprehensively evaluate the proposed method, we compare DSTD against a strong set of state-of-the-art baselines from two categories: probabilistic methods and deterministic methods. We include recent diffusion-based models designed for time-series or spatio-temporal forecasting, such as TimeGrad (Rasul et al., 2021), DiffSTG (Wen et al., 2023), and USTD (Hu et al., 2024). For deterministic baselines, we consider DCRNN (Li et al., 2018), Graph Wavenet (Wu et al., 2019b), GMAN (Zheng et al., 2020), STEP (Shao et al., 2022), and DGCRN (Li et al., 2023a). A detailed description of each baseline is available in Appendix B.1.

**Evaluation Metrics**  We assess the performance of all models using three standard metrics. To evaluate the point forecasts, the median of the predicted distribution for probabilistic models, we use Mean Absolute Error (MAE) and Root Mean Squared Error (RMSE). To evaluate the quality of the full predicted probability distribution, we use the Continuous Ranked Probability Score (CRPS).

### 4.2 RESULTS

Table 2 presents the forecasting results, comparing DSTD against the baselines on the METR-LA and PEMS-BAY datasets. The results provide strong empirical evidence that our decoupled architecture, DSTD, not only sets a new state-of-the-art among probabilistic models but also achieves highly competitive or even superior performance compared to top deterministic methods. This demonstrates that the efficiency gains from our design do not come at the cost of expressive power.

On the PEMS-BAY dataset, the superiority of DSTD is particularly evident. It achieves the best performance across all three forecasting horizons, outperforming all baselines, including the strongest deterministic models. Specifically, it consistently obtains the best CRPS scores, indicating that

Table 2: Performance comparison of the deterministic and probabilistic methods. **Bold** denotes the best result and underline denotes the best-performing method within the other category.

| | Method | 15 Min | | | 30 Min | | | 60 Min | | |
|---|---|---|---|---|---|---|---|---|---|---|
| | | MAE | RMSE | CRPS | MAE | RMSE | CRPS | MAE | RMSE | CRPS |
| METR-LA | DCRNN | 2.77 | 5.38 | - | 3.15 | 6.45 | - | 3.60 | 7.60 | - |
| | Graph WaveNet | 2.69 | 5.15 | - | 3.07 | 6.22 | - | 3.53 | 7.37 | - |
| | GMAN | 2.80 | 5.55 | - | 3.12 | 6.49 | - | 3.44 | 7.35 | - |
| | STEP | **2.61** | **4.98** | - | 2.96 | 5.97 | - | 3.37 | 6.99 | - |
| | DGCRN | 2.62 | 5.01 | - | 2.99 | 6.05 | - | 3.44 | 7.19 | - |
| | TimeGrad | 3.94 | 8.39 | 0.085 | 4.57 | 10.05 | 0.101 | 5.04 | 10.50 | 0.132 |
| | DiffSTG | 3.06 | 6.54 | 0.062 | 3.47 | 7.60 | 0.081 | 4.18 | 8.72 | 0.112 |
| | USTD | 3.47 | 7.69 | **0.051** | 4.11 | 9.15 | **0.061** | 5.03 | 10.91 | **0.073** |
| | DSTD | 2.65 | 5.55 | 0.094 | **2.88** | **5.95** | 0.138 | **3.22** | **6.77** | 0.165 |
| PEMS-BAY | DCRNN | 1.38 | 2.95 | - | 1.74 | 3.97 | - | 2.07 | 4.74 | - |
| | Graph WaveNet | 1.30 | 2.74 | - | 1.63 | 3.70 | - | 1.95 | 4.52 | - |
| | GMAN | 1.34 | 2.82 | - | 1.62 | 3.72 | - | 1.86 | 4.32 | - |
| | STEP | 1.26 | 2.73 | - | 1.55 | 3.58 | - | 1.79 | 4.20 | - |
| | DGCRN | 1.28 | 2.69 | - | 1.59 | 3.63 | - | 1.89 | 4.42 | - |
| | TimeGrad | 2.27 | 4.74 | 0.030 | 2.35 | 4.92 | 0.030 | 2.49 | 5.26 | 0.032 |
| | DiffSTG | 1.19 | 2.38 | 0.015 | 1.44 | 3.11 | 0.018 | 1.78 | 4.12 | 0.023 |
| | USTD | 1.41 | 2.51 | 0.019 | 1.46 | **2.97** | 0.020 | 1.86 | **3.81** | 0.025 |
| | DSTD | **1.14** | **2.29** | **0.014** | **1.37** | 2.99 | **0.017** | **1.70** | 3.91 | **0.021** |

it produces the highest quality probabilistic forecasts. On the more complex METR-LA dataset, DSTD continues to show exceptional performance. It is the clear top performer among all probabilistic models in terms of point forecasting. More impressively, it directly competes with and often surpasses the best deterministic models. Although STEP shows a slight edge in short-term (15 min) forecasting, the results of DSTD are remarkably close (e.g., an MAE of 2.65 vs. STEP's 2.61). As the forecast horizon extends to 30 and 60 minutes, DSTD achieves the best overall MAE and RMSE, demonstrating its robustness for longer-term predictions.

These results validate our design principle. The success of DSTD shows that by decoupling propagation and transformation, it is possible to build a diffusion model that achieves state-of-the-art generative quality without sacrificing point-forecast accuracy. The ability to outperform specialized deterministic models on several key metrics stems from two critical advantages of probabilistic modeling. First, the denoising training objective acts as a powerful regularizer, preventing the model from overfitting to superficial data patterns and promoting robustness. Second, the capacity of DSTD to model the uncertain nature of real-world spatio-temporal data allows it to better estimate the true central tendency of the distribution. Contrary to deterministic models, which are compelled toward a blurry average prediction to minimize squared error across potential outcomes, the median prediction derived from DSTD is often more accurately aligned with a realistic outcome. The ability to achieve this while providing rich and uncertain forecasts underscores the significant advantages of the proposed approach.

### 4.3 ANALYSIS OF COMPUTATIONAL EFFICIENCY

We conduct 12-step forecasting experiments on the METR-LA dataset to compare the efficiency of DSTD against other diffusion-based baselines. Table 3 presents a comprehensive analysis, comparing not only practical inference speed but also training time, theoretical computational complexity of the denoising block, and the total number of parameters. The results reveal a dramatic improvement in computational efficiency. Specifically, to generate 32 samples, DSTD requires only 0.63 seconds, which is approximately 8.4 times faster than USTD and over 20 times faster than DiffSTG. This remarkable speed-up is a direct consequence of our core design principle. The theoretical complexity analysis explains the fundamental source of this efficiency. DiffSTG is constrained by a costly $\mathcal{O}(N \cdot H^2 \cdot T^2)$ complexity, scaling quadratically with the hidden dimension $H$. USTD relies on a complex attention-based encoder, leading to an $\mathcal{O}(N^2 \cdot H)$ complexity that scales quadratically with the number of nodes $N$. By decoupling the architecture into parameter-free propagation layers

Table 3: Computational efficiency comparison of diffusion-based models on the METR-LA dataset. Complexity indicates the theoretical computational complexity of a single denoising block. $N$, $H$, and $T$ are the number of nodes, hidden dimension size, and time window, respectively. # denotes the number of. Training times are reported in seconds per epoch. Training time for USTD is not directly comparable, as it requires a 2-stage training process. Inference times are reported in seconds for generating a varied number of samples ($S$).

| Method | Complexity | # Parameters | Training Time | $S = 8$ | $S = 16$ | $S = 32$ |
|--------|------------|--------------|---------------|---------|----------|----------|
| DiffSTG | $\mathcal{O}(N \cdot H^2 \cdot T^2)$ | 1.039M | 172.70 | 3.31 | 6.48 | 13.00 |
| USTD | $\mathcal{O}(N^2 \cdot H)$ | 0.471M | - | 1.91 | 2.86 | 5.31 |
| DSTD | $\mathcal{O}(N \cdot H \cdot T^2)$ | 0.438M | 36.24 | **0.27** | **0.36** | **0.63** |

and lightweight transformation layers, DSTD avoids the expensive computations, achieving a highly efficient $\mathcal{O}(N \cdot H \cdot T^2)$ complexity that scales linearly with both $N$ and $H$. This comprehensive efficiency extends to training speed and memory footprint. DSTD is significantly faster to train, approximately 4.76 times faster than DiffSTG. A direct comparison with USTD is not applicable due to its 2-stage training process. Furthermore, DSTD is the most lightweight model, requiring only 0.438M parameters, compared to 0.471M for USTD and 1.039M for DiffSTG. These combined results empirically and theoretically confirm that DSTD successfully resolves the critical efficiency problem, establishing the decoupled approach as a practical and scalable solution.

## 4.4 ABLATION STUDY

Table 4: Ablation study on the METR-LA dataset. We compare the full DSTD against variants that individually remove the position-aware embedding (- Pos. Emb.), the decoupling principle (- Decoupling), the linearity principle (- Linearity), and the dynamic multi-scale aggregation (- Dyn. Agg.). Coupled ST-GCN represents a baseline that does not incorporate any proposed principles.

| Method | MAE | RMSE | CRPS | Time |
|--------|-----|------|------|------|
| DSTD | **3.22** | **6.77** | 0.165 | **0.63** |
| - Pos. Emb. | 3.26 | 6.92 | 0.164 | 0.64 |
| - Decoupling | 3.25 | 6.83 | **0.152** | 0.74 |
| - Linearity | 3.37 | 7.37 | 0.167 | 0.67 |
| - Dyn. Agg. | 3.36 | 7.20 | 0.160 | 0.73 |
| Coupled ST-GCN | 3.40 | 7.49 | 0.164 | 0.78 |

To empirically validate the core design principles of DSTD, we conduct a comprehensive ablation study on the METR-LA dataset for a 12-step forecasting horizon. The results are presented in Table 4. The most critical comparison is between our full model and the Coupled ST-GCN backbone, which uses none of the proposed principles. DSTD significantly outperforms this complex baseline across most metrics while being 20% faster, demonstrating that our principled simplifications collectively lead to a more effective and efficient model. Analyzing each principle individually confirms its importance. Removing decoupling (by re-introducing learnable parameters) results in only a minor degradation in MAE but leads to a considerable increase in computational time. Furthermore, the observation that it achieves a numerically sharper CRPS while having worse MAE is critical. This trade-off suggests that re-introducing coupled learning makes the model prone to over-confidence and less generalizable. Conversely, the decoupling principle allows DSTD to maintain stability, acting as a strong regularizer that achieves a superior balance between efficiency, high point-forecast accuracy, and robust distributional quality. Removing linearity (by adding ReLU activations) or dynamic multi-scale aggregation leads to much larger degradation, validating that these components operate synergistically with the decoupling framework. Finally, the removal of the position-aware embedding also results in a slight performance drop, confirming its role as a useful inductive bias.

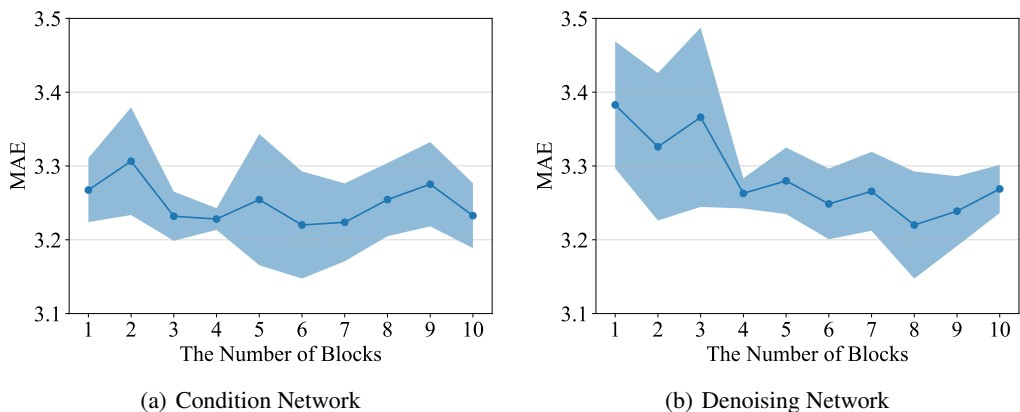

(a) Condition Network        (b) Denoising Network

Figure 2: Performance analysis (MAE) with respect to the number of blocks in (a) the condition network and (b) the denoising network on the METR-LA dataset. The line and shaded area represent the mean and standard deviation, respectively.

### 4.5 ANALYSIS OF ROBUSTNESS AND SCALABILITY

We evaluate the performance of DSTD by varying the depth of the condition network and the denoising network to analyze structural robustness. Figure 2 reveals a typical pattern for GNNs. Performance improves as the number of blocks increases, effectively capturing wider spatial dependencies. However, after an optimal point, performance begins to degrade as an excessively large receptive field incorporates irrelevant noise. This analysis demonstrates that DSTD has a predictable performance curve and confirms the existence of a stable, optimal architectural depth.

We conduct a long-term forecasting experiment by extending the prediction horizon to 120 minutes (24 steps). Table 5 shows that DSTD outperforms the strongest deterministic (STEP) and probabilistic (Diff-STG) baselines on MAE and RMSE, while its CRPS score remains highly competitive. We visualize forecasts of DSTD and DiffSTG in Figure 3. The visualizations reveal that DSTD generates more stable predictions that better track the evolving ground-truth trends, avoiding the abrupt spikes occasionally produced by DiffSTG (e.g., Figure 3(f)). Furthermore, DSTD consistently produces sharper prediction intervals, indicating more confident estimates. These analyses demonstrate the robustness and scalability of DSTD.

Table 5: Performance comparison on a long-term forecasting task (120-minute horizon) on the METR-LA dataset.

| Method | MAE | 120 Min RMSE | CRPS |
|---|---|---|---|
| STEP | 5.78 | 9.71 | - |
| DiffSTG | 4.78 | 9.27 | **0.160** |
| DSTD | **3.74** | **8.06** | 0.165 |

## 5 CONCLUSION

In this paper, we address the critical computational bottleneck that hinders the practical application of diffusion models for spatio-temporal graph forecasting. We propose an architectural design principle: the decoupling of information propagation and feature transformation. We instantiate this principle in a highly efficient framework, DSTD, which features a concise, linear design and a dynamic multi-scale aggregation mechanism. Through extensive experiments on widely-used benchmarks, we demonstrate that DSTD achieves state-of-the-art or highly competitive performance against both probabilistic and deterministic baselines, while offering a dramatic reduction in inference time. Our analyses of the architectural design, robustness, and scalability to challenging long-horizon tasks further validate our principled simplification as a powerful and reliable solution. We acknowledge that this work operates under the assumption of a static graph structure, a limitation common in current benchmarks. Future work can focus on extending DSTD to incorpo-

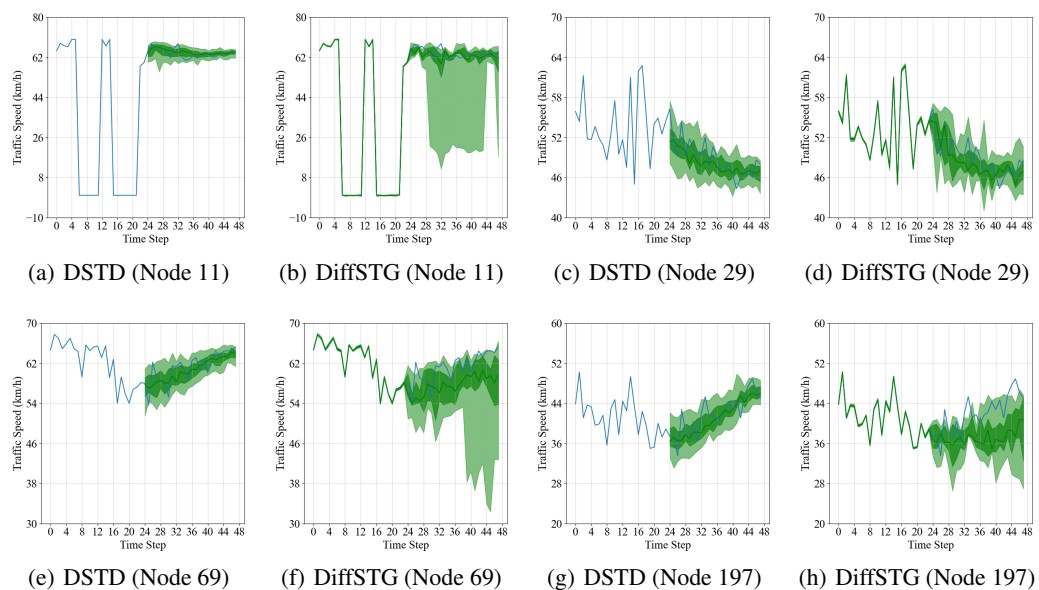

Figure 3: Visualization of long-term forecasts, comparing DSTD against DiffSTG on the METR-LA dataset. The blue line represents the ground truth, the green line is the predicted median, and the dark and light shaded areas correspond to the 50% and 90% prediction intervals, respectively.

rate dynamic graph learning mechanisms, addressing the evolving spatial relationships inherent in real-world spatio-temporal systems.

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

## A    RELATED WORK

### A.1    SPATIO-TEMPORAL GRAPH FORECASTING

Spatio-temporal graph forecasting is a crucial task in various real-world applications such as traffic prediction (Yu et al., 2018; Jiang et al., 2023), weather forecasting (Han et al., 2021; Ji et al., 2025), and energy demand estimation (Sharma et al., 2022; Peng et al., 2024). Early deep learning approaches successfully capture complex dependencies by combining Graph Neural Networks (GNNs) with sequence models. For instance, DCRNN (Li et al., 2018) integrates diffusion convolution with recurrent neural networks, whereas Graph WaveNet (Wu et al., 2019b) combines GNNs with temporal convolutional networks. Subsequent works incorporate attention mechanisms (Zheng et al., 2020) and transformers (Zhang et al., 2023) to better capture long-range dependencies. More recent studies explore dynamic graph structures (Shang et al., 2021; Li et al., 2023b) and pre-trained models (Shao et al., 2022; Fang et al., 2024). Although these models have progressively advanced the state-of-the-art, the vast majority focus on deterministic point forecasting, failing to capture the inherent uncertainty in real-world spatio-temporal graph data.

### A.2    SPATIO-TEMPORAL DIFFUSION MODELS

Denoising diffusion models (Ho et al., 2020; Song et al., 2021) have recently emerged as a powerful class of generative models, initially demonstrating remarkable success in the image (Rombach et al., 2022a; Podell et al., 2024) and audio (Kong et al., 2021; Huang et al., 2022) domains. Their ability to model complex data distributions leads to their adoption in time-series applications for general-purpose probabilistic forecasting (Rasul et al., 2021) and imputation (Tashiro et al., 2021). This trend naturally extends to spatio-temporal graph data, where capturing uncertainty is crucial. Models, such as DiffSTG (Wen et al., 2023) and USTD (Hu et al., 2024), are among the first to apply diffusion models specifically to spatio-temporal graph forecasting, showing promising results in generating probabilistic predictions. However, these pioneering works rely on complex GNN architectures within their iterative denoising process, inheriting the significant computational overhead that makes them slow and difficult to scale.

### A.3    DECOUPLED GRAPH NEURAL NETWORKS

A line of research inspires the decoupling principle, which focuses on improving the efficiency and performance of GNNs through simplification. As GNNs become deeper and more complex, they often suffer from issues such as over-smoothing, over-fitting, and high computational costs (Waikhom & Patgiri, 2023). SGC (Wu et al., 2019a) is a pioneering work that showed removing non-linearities and collapsing weight matrices can yield an effective linear model. LightGCN (He et al., 2020) further advances this philosophy, which demonstrates that removing both non-linearities and feature transformations from the propagation process leads to state-of-the-art results in collaborative filtering. This idea of separating the information propagation step from the feature transformation step is also shared by other models, such as APPNP (Gasteiger et al., 2019), SIGN (Frasca et al., 2020), and TFE-GNN (Duan et al., 2024), which decouple propagation from the neural network to learn larger neighborhood contexts. Our work is the first to recognize the profound potential of this decoupling principle to overcome the computational bottleneck in graph-based diffusion models. By integrating this principled simplification into a diffusion framework, DSTD uniquely combines the generative power of diffusion models with the efficiency of modern lightweight GNN architectures.

## B    EXPERIMENTAL DETAILS

### B.1    BASELINES

We adopt the following diffusion-based models as baselines for probabilistic forecasting:

- **TimeGrad** Rasul et al. (2021), a diffusion-based autoregressive time-series forecasting model that uses RNN to encode history.
- **DiffSTG** Wen et al. (2023), a diffusion-based STG forecasting model incorporating UGNet to capture spatial dependencies.

- **USTD** Hu et al. (2024), a diffusion-based STG multitask model unifying forecasting and kriging using a pre-trained spatio-temporal encoder and task-specific denoising decoders.

We employ the following representative state-of-the-art deterministic models for comparison:

- **DCRNN** Li et al. (2018), combining a graph diffusion convolution with RNNs to capture spatial and temporal dependencies.
- **Graph WaveNet** Wu et al. (2019b), integrating GCNs with TCNs to enhance effectiveness in capturing temporal dependencies.
- **GMAN** Zheng et al. (2020), utilizing spatial and temporal attention mechanisms to model dynamic spatial and temporal correlations.
- **STEP** Shao et al. (2022), leveraging an unsupervised pre-trained time series transformer to learn long-term history.
- **DGCRN** Li et al. (2023a), evolving graph structures using hyper-networks to extract dynamic characteristics from node attributes.

## B.2 IMPLEMENTATION DETAILS

We evaluate the models on three forecasting horizons: 15, 30, and 60 minutes, corresponding to 3, 6, and 12 time steps, respectively. The history length is set to be equal to the prediction horizon (e.g., 3 steps of history are used to predict 3 steps into the future). We follow the standard data processing protocol, partitioning the datasets into training, validation, and testing sets with a 7:1:2 chronological split (Li et al., 2018; Shang et al., 2021). We utilize multi-layer perceptrons for all projection functions. We train DSTD using the EMA strategy (Lee et al., 2024) and the Adam optimizer with a batch size of 64, a learning rate of 0.001, and a weight decay of 1e-6. We set the hidden dimension to 32, with 6 condition blocks and 8 denoising blocks. The total number of diffusion steps is $K = 200$, and we use $M = 40$ steps for accelerated inference. We adopt a quadratic noise schedule, where $\beta_k = \left( \frac{K-k}{K-1} \sqrt{\beta_1} + \frac{k-1}{K-1} \sqrt{\beta_K} \right)^2$ with $\beta_1 = 0.0001$ and $\beta_K = 0.1$. The hyperparameters for all baseline models are set according to their original papers. The reported results are the average of five independent runs.

## C  ANALYSIS OF DYNAMIC MULTI-SCALE AGGREGATION

### C.1  VISUALIZATION OF LEARNED AGGREGATION STRATEGIES

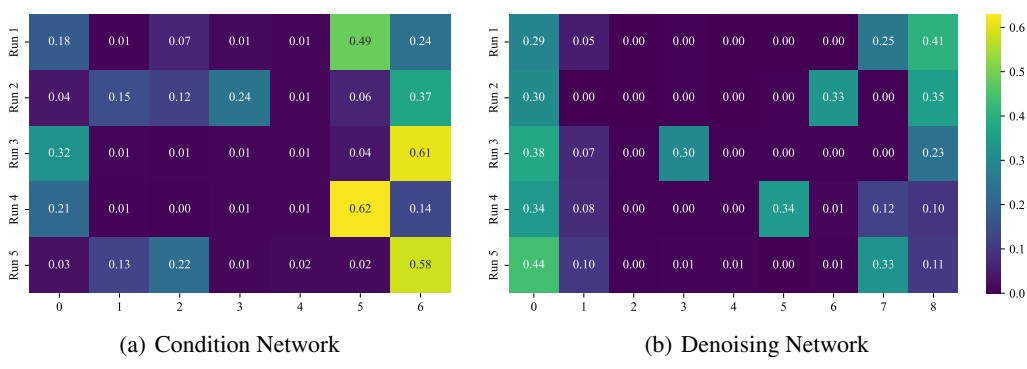

(a) Condition Network            (b) Denoising Network

Figure 4: Visualization of the learned attention weights from the dynamic multi-scale aggregation mechanism for (a) the condition network and (b) the denoising network on the METR-LA dataset. Each row represents an independent run, and the x-axis represents the block depth.

To understand how DSTD utilizes features from different depths, we visualize the learned attention weights ($\alpha^{(b)}$) of the dynamic multi-scale aggregation mechanism in Figure 4. The heatmaps, generated from a 12-step forecast on the METR-LA dataset, show the learned layer importance

for the condition and denoising networks across five independent runs. The results reveal that the two networks learn distinct strategies. The condition network consistently concentrates its attention on deeper blocks (specifically 5 and 6), prioritizing a larger receptive field to encode the historical context. In contrast, the denoising network learns a more diverse strategy, always assigning high importance to the initial representation while utilizing a mixture of embeddings from multiple scales. This suggests the generative process requires both highly local and global information.

## C.2 MULTI-HEAD EXTENSION

The diverse strategies learned by the denoising network motivate us to explore whether a more complex, multi-head aggregation mechanism could further enhance performance. We replace the single aggregation mechanism with a multi-head version and train on the METR-LA dataset for a 12-step forecasting horizon.

Table 6: Performance comparison between the standard DSTD (Single-head) architecture and an extended DSTD (Multi-head) version on the METR-LA dataset.

| Method | MAE | RMSE | CRPS | Time |
|---|---|---|---|---|
| DSTD (Single-head) | **3.22** | **6.77** | 0.165 | **0.63** |
| DSTD (Multi-head) | 3.34 | 7.17 | **0.163** | 0.71 |

Table 6 shows that the multi-head variant performs worse than the single-head DSTD. This suggests that learning multiple complex aggregation strategies is not only computationally complex but also less effective than the well-optimized strategy learned by DSTD. This finding further reinforces the core philosophy of the paper that principled simplification is a powerful and effective.

## D ARCHITECTURAL ANALYSIS OF DSTD AND SIMPLE LINEAR FILTERS

A critical question regarding the DSTD framework is whether its performance stems from its specific architectural choices, namely, the decoupled block design, or if a standard simplified GNN (e.g., LightGCN) could serve as a sufficient backbone. Specifically, it is important to verify whether the linear design of DSTD is equivalent to a simple linear filter, potentially sacrificing the representational capacity for efficiency. To investigate this, we design an experiment on the METR-LA dataset for a 12-step forecasting horizon to isolate the contribution of the architectural components. We construct LightGCN variant to mimic a simple linear filter backbone. This model is identical to DSTD except that we completely remove the $1 \times 1$ convolution layer from all denoising blocks. This modification eliminates the mechanism for learning temporal correlations within the denoising process, retaining only the parameter-free spatial propagation.

Table 7 demonstrates a significant performance gap. LightGCN variant shows a catastrophic degradation in predictive accuracy. These findings indicate that simply applying diffusion to a standard simplified GNN backbone is insufficient for complex spatio-temporal forecasting. It confirms that the performance of DSTD critically depends on its unique architecture, which intelligently interleaves propagation and transformation, rather than merely the principle of simplification. Furthermore, the failure of LightGCN variant suggests that $1 \times 1$ convolution is an essential component for capturing temporal dynamics. The design of DSTD achieves high efficiency without sacrificing the representational power required for the task.

Table 7: Performance comparison between DSTD and LightGCN variant on the METR-LA dataset.

| Method | MAE | RMSE | CRPS |
|---|---|---|---|
| DSTD | 3.22 | 6.77 | 0.165 |
| LightGCN | 5.81 | 12.32 | 0.152 |

We observe an interesting phenomenon regarding the CRPS metric. LightGCN variant achieves a numerically lower CRPS of 0.152 compared to DSTD. This suggests that a model can produce overconfident, sharper distributions even when its central prediction is fundamentally inaccurate. This finding highlights that DSTD achieves a superior and more robust balance between point-forecast and generative quality, avoiding the pitfall of over-confidence seen in the simplified variant.

# E  VISUALIZATIONS

To provide a more comprehensive qualitative assessment of DSTD, we present additional visualizations of DSTD forecasting. Figure 5 and 6 illustrate the predictions for the 60-minute (12-step) forecasting horizon, for the first 16 nodes from the test set of the METR-LA dataset and PEMS-BAY dataset, respectively.

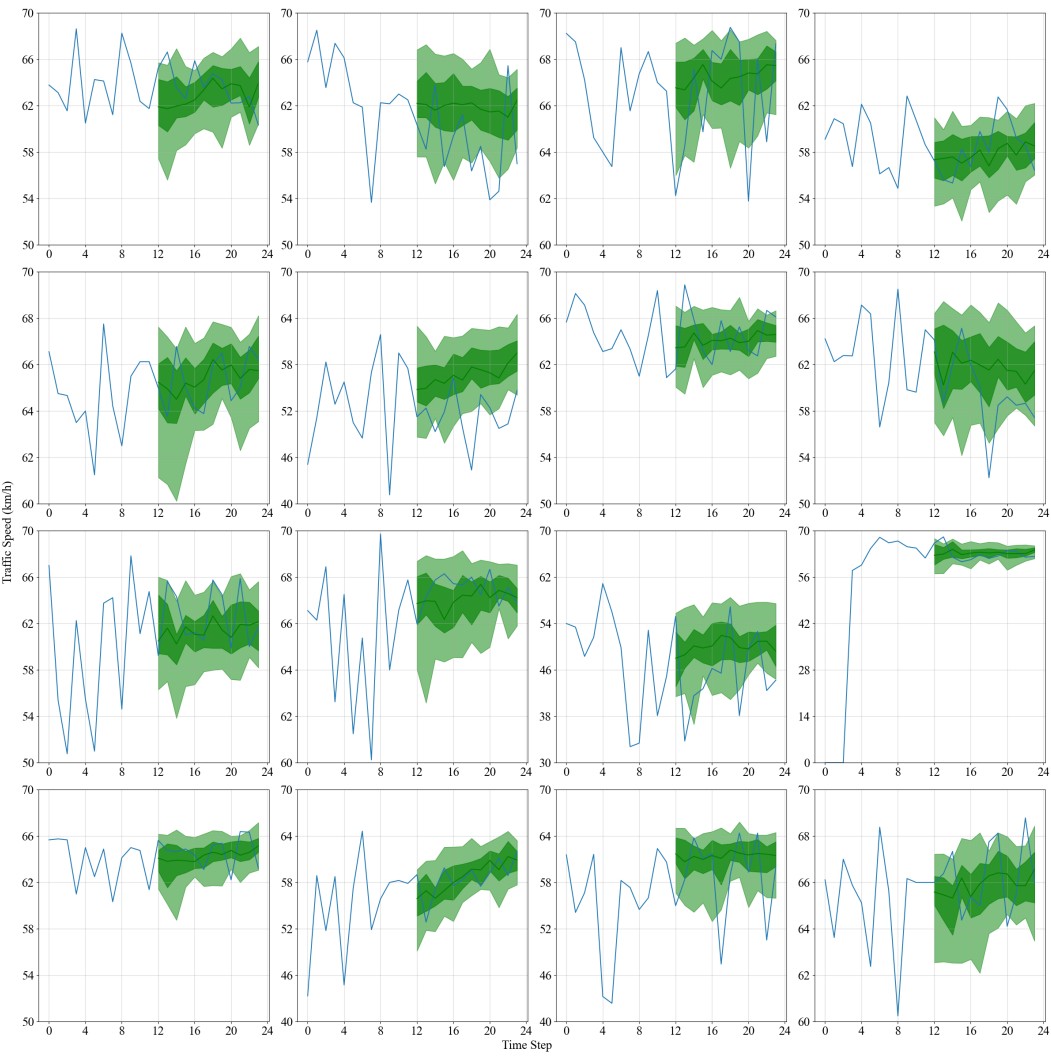

Figure 5: Visualizations of 60-minute (12-step) forecasts by DSTD on the first 16 nodes from the test set of the METR-LA dataset. The blue line represents the ground truth, the green line indicates the predicted median, and the dark and light shaded areas correspond to the 50% and 90% prediction intervals, respectively.

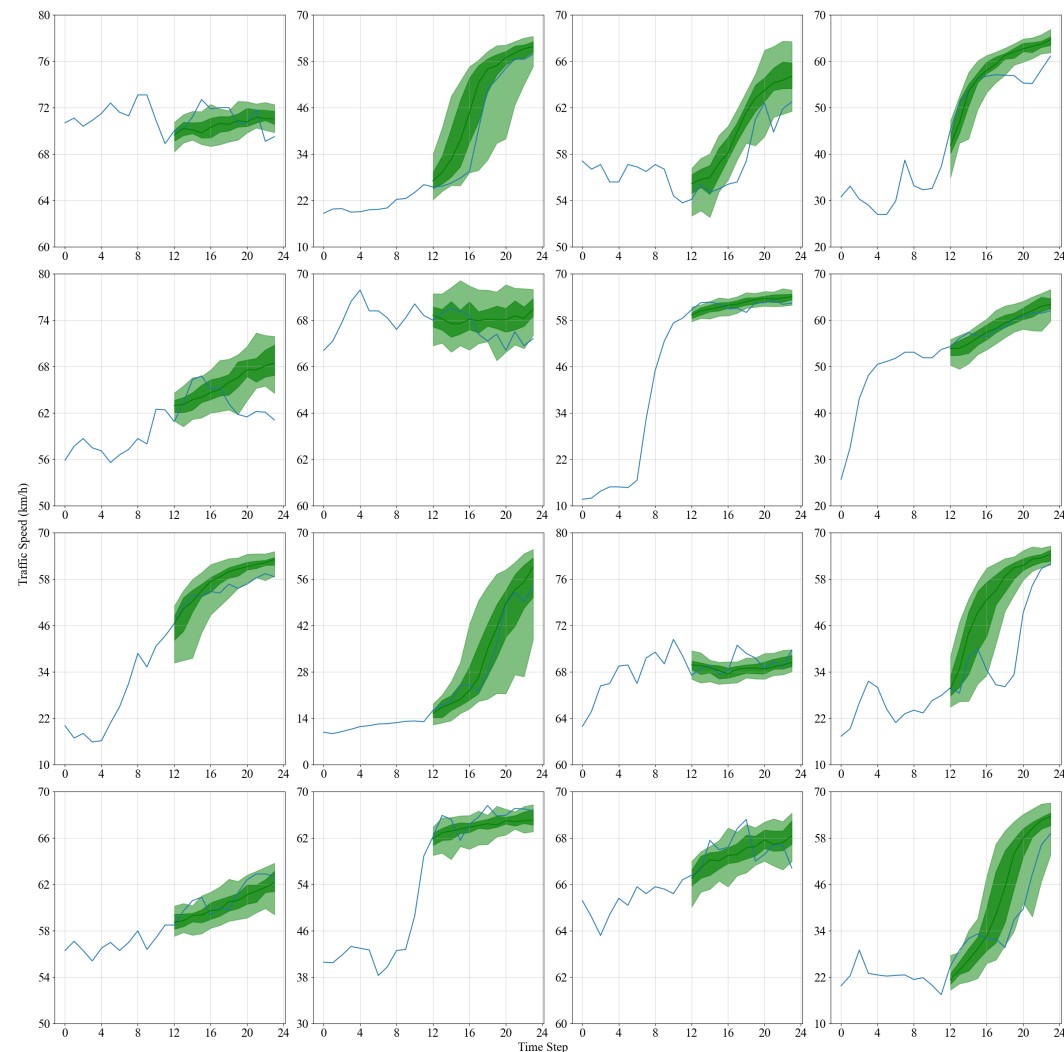

Figure 6: Visualizations of 60-minute (12-step) forecasts by DSTD on the first 16 nodes from the test set of the PEMS-BAY dataset. The blue line represents the ground truth, the green line indicates the predicted median, and the dark and light shaded areas correspond to the 50% and 90% prediction intervals, respectively.

## F    LLM USAGE

We use a large language model (LLM) as an assistive tool in preparing this manuscript. The primary role of LLM is to help refine the narrative and to polish the writing for clarity. All scientific claims, experimental results, and final text are directed, verified, and written by the authors, who are responsible for the content of this paper.

