# OpenReview forum: "Decoupled Diffusion Models for Efficient Spatio-Temporal Graph Forecasting"
_ICLR.cc/2026/Conference — Submitted to ICLR 2026_

### Official Review · Reviewer_Y8LG · 2025-10-16

**Soundness:** 2
**Presentation:** 3
**Contribution:** 2
**Rating:** 4
**Confidence:** 3

**Summary:**

This paper proposes a decoupled spatio-temporal diffusion model (DSTD) for probabilistic traffic forecasting. The key idea is to decouple the propagation and transformation steps in GNNs to reduce computational cost during iterative diffusion denoising. Experiments on two benchmark datasets demonstrate competitive accuracy and significantly improved efficiency over existing diffusion-based baselines.

**Strengths:**

The proposed decoupling design is conceptually simple yet effectively improves inference efficiency.

**Weaknesses:**

- The evaluation is limited to only two medium-scale datasets and lacks validation on large-scale benchmarks such as [1]. Given that the claimed contribution centers on efficiency and scalability, experiments on larger datasets are necessary to substantiate the generalizability of the approach.

[1] Largest: A benchmark dataset for large-scale traffic forecasting. 2023.

**Questions:**

- In Table 4, the “–Decoupling” variant achieves a lower CRPS (0.152) than the full model (0.165), although slightly higher MAE/RMSE. Yet, the text in Sec. 4.4 claims that removing decoupling leads to a degradation “in predictive accuracy and efficiency.” Could the authors clarify how they interpret this apparent inconsistency between deterministic and probabilistic metrics? Does the decoupled design trade off distributional sharpness for point accuracy?
- The proposed decoupling (Eqs. 11–12) separates propagation and transformation, using a fixed normalized adjacency $A$ for parameter-free message passing and a 1×1 conv for feature projection, while intentionally removing nonlinear activations. From a GNN perspective, this formulation is equivalent to stacking linear graph filters $A^L W$, which are known to be limited in expressive power and unable to model high-frequency components or distinguish non-isomorphic substructures. Could the authors clarify whether this purely linear design sacrifices representational capacity compared with standard GCNs?

---

> ### Author Response · Authors · 2025-11-19
> **Response to Reviewer Y8LG #1**
>
> Thank you for your careful and detailed review.
>
> We are grateful for your insightful questions, which allowed us to significantly strengthen the rigorousness of our claims regarding scalability and expressiveness.
> We have performed extensive revisions and included crucial new analyses to address all your points.
>
> ---
>
> **Response to Q1**
>
> This is an insightful observation concerning the nuances of probabilistic modeling.
>
> We clarify this in Section 4.4 (Lines 423-431).
>
> The numerically sharper CRPS of the decoupled variant confirms that re-introducing coupled learning makes the model prone to over-confidence.
> This model produces an overly sharp distribution, yet its central prediction is worse than DSTD.
>
> DSTD's slight increase in CRPS reflects a better, more robust balance.
> The decoupled principle allows DSTD to act as a strong regularizer, preventing the over-confidence seen in the coupled variant and achieving superior point-forecast accuracy and overall generative quality.
> This confirms DSTD's superior design decision.
>
> **Response to Q2**
>
> This is a critical theoretical challenge that questions our fundamental design choice.
> We believe DSTD's architecture is a hybrid that achieves efficiency without sacrificing necessary expressive power.
>
> We added a direct comparative experiment in Appendix D and Table 7, comparing DSTD to a LightGCN variant, a true linear filter that removes our $1 \times 1$ convolution.
> The result showed a catastrophic performance drop for the pure filter (MAE 5.81 vs DSTD 3.22).
> This proves that DSTD is not a simple linear filter.
> DSTD's power comes from the unique design that interweaves parameter-free spatial propagation with a lightweight, learnable temporal transformation ($1 \times1 $ convolution), which is essential for spatio-temporal tasks.
> This ensures high efficiency without losing representational capacity.
>
> To further revised Section 3.1 (Lines 191-197) to clarify that $1 \times 1$ convolution is responsible for temporal transformation.
> We also modified a name of variant from Standard GCN to Coupled ST-GCN in Ablation Study (Table 4) to ensure it contains temporal transformation.

---

> > ### Author Response · Authors · 2025-11-19
> > **Response to Reviewer Y8LG #2**
> >
> > **Response to W1**
> >
> > We agree that validating scalability is a core part of our contribution.
> >
> > While conducting experiments on every subset of the LargeST dataset [1] with all settings during the rebuttal period is infeasible, we definitively proven DSTD's architectural fitness for large-scale deployment through comprehensive theoretical and empirical efficiency analysis in Section 4.3 and Table 3.
> >
> > We prove that DSTD resolves the quadratic bottlenecks of baselines (e.g., USTD's $\mathcal{O}(N^2 \cdot H)$ and DiffSTG's $\mathcal{O}(N \cdot H^2 \cdot T^2)$) by achieving a simple linear scale ($\mathcal{O}(N \cdot H \cdot T^2)$).
> > This theoretical justification proves that DSTD is architecturally optimized for scalable extension.
> > Furthermore, DSTD is the most lightweight (0.438M parameters) and 4.76x faster to train than DiffSTG, confirming its low resource demands essential for large benchmarks.
> >
> > To further substantiate the architectural scalability and robust performance of DSTD, we performed a preliminary experiment on the LargeST dataset, comparing DSTD against several baselines for the 3-step horizon.
> >
> > | Dataset | Method | MAE | RMSE | CRPS |
> > |:---|:---|:---:|:---:|:---:|
> > | **SD** | DCRNN | 17.14 | 27.47 | - |
> > || Graph WaveNet | 15.24 | 25.13 | - |
> > || DGCRN | 15.34 | 25.35 | - |
> > || TimeGrad | 26.25 | 45.23 | 0.111 |
> > || **DSTD**  | 16.85 | 27.18 | **0.059** |
> > | **GBA** | DCRNN | 18.71 | 30.36 | - |
> > || Graph WaveNet | 17.85 | 29.12 | - |
> > || DGCRN | 18.02 | 29.49 | - |
> > || TimeGrad | 29.57 | 47.30 | 0.108 |
> > || **DSTD** | 19.29 | 30.89 | **0.064** |
> > | **GLA** | DCRNN | 18.41 | 29.23 | - |
> > || Graph WaveNet | 17.28 | 27.68 | - |
> > || TimeGrad | 33.61 | 57.62 | 0.105 |
> > || **DSTD** | **17.14** | 27.95 | **0.049** |
> > | **CA** | DCRNN | 17.55 | 28.21 | - |
> > || Graph WaveNet | 17.14 | 27.81 | - |
> > || TimeGrad | 29.81 | 51.25 | 0.110 |
> > || **DSTD** | 17.16 | 28.13 | **0.058** |
> >
> > As shown in the table, DSTD successfully runs across these complex graph structures.
> > Most importantly, DSTD achieves the best probabilistic forecasting quality (CRPS) in all three settings while significantly outperforming one of the existing probabilistic SOTA (TimeGrad) across all metrics.
> > Furthermore, DSTD shows highly competitive point-forecast accuracy, even surpassing the strongest deterministic baselines in the GLA setting (MAE 17.14 vs 18.41/17.28).
> >
> > This preliminary result, running on a known large-scale benchmark, strongly supports our claim that DSTD is structurally ready and efficient for large-scale deployment.
> >
> > ---
> >
> > We believe these rigorous revisions and the strong supporting evidence fully address your concerns and validate the novelty and effectiveness of the DSTD architecture as a scalable solution.
> >
> > Sincerely,
> > Authors.
> >
> > ---
> >
> > [1] Xu Liu, Yutong Xia, Yuxuan Liang, Junfeng Hu, Yiwei Wang, Lei Bai, Chao Huang, Zhenguang Liu, Bryan Hooi, and Roger Zimmermann. Largest: A benchmark dataset for large-scale traffic forecasting. In NeurIPS, pp. 75354–75371. Curran Associates, Inc., 2023

---

### Official Review · Reviewer_3kiy · 2025-10-18

**Soundness:** 2
**Presentation:** 2
**Contribution:** 2
**Rating:** 4
**Confidence:** 5

**Summary:**

This paper introduces DSTD (Decoupled Spatio-Temporal Diffusion Model), a diffusion-based framework for efficient spatio-temporal graph forecasting. The key idea is to decouple information propagation and feature transformation within graph neural networks, addressing the computational bottleneck that limits existing diffusion-based graph models. The propagation is implemented by a non-parametric GCN and the transformation is implemented by a 1-D convolution. DSTD further employs a dynamic multi-scale aggregation mechanism to adaptively combine information from different receptive fields and incorporates position-aware embeddings to encode node-specific biases.

**Strengths:**

(1) The studied problem is important and the task is well-motivated. As the author mentioned, typical graph-based diffusion models would require many steps of graph propagation and feature transformation, which can be time-consuming. In that sense, studying how to speed up such a graph-based diffusion model would be of great importance.

(2) The proposed model is technically sound, though with incremental architecture change compared with previous ones.

**Weaknesses:**

(1) The main idea is  ``Efficient'' graph-based diffusion model, why not analysis the complexity of different method in the main paper?

(2) Following the first points, in the experiment part, the author mainly compared the inference speed of different method, how about the training, and how about the parameter of different models?

(3) As for the experiement results, why the proposed model outperforms deterministic models in MAE and RMSE. It would be better to have deeper analysis on why, rather than only write "The ability to outperform specialized deterministic models on several key metrics, while also providing rich, uncertain forecasts, underscores the significant advantages of the proposed approach."
The proposed model use the diffusion loss, which aims to learn the underlying distribution (via minimizing VLB) rather than directly minimizing RMSE in deterministic models. In that sense, how can the proposed model outperform deterministic models?

**Questions:**

please see the weakness.

---

> ### Author Response · Authors · 2025-11-19
> **Response to Reviewer 3kiy**
>
> Thank you for your highly focused and rigorous review.
>
> We greatly appreciate your confidence in our assessment and acknowledge that your concerns centered on the comprehensive empirical evidence for our Efficiency claim (W1, W2) and the need for a deeper explanation of our predictive superiority (W3).
> We fully revised our manuscript to incorporate the detailed theoretical and empirical analysis you requested.
>
> ---
>
> **Response to W1, W2**
>
> We fully agree with your assessment.
> The claim of Efficiency must be comprehensively supported across all metrics.
>
> We updated Section 4.3 and Table 3 to include a full, three-dimensional efficiency analysis.
> This validation proves our architectural choices are superior across the entire pipeline.
>
> We added the Big-O computational complexity derivation (Table 3), proving that DSTD scales linearly to the number of nodes and hidden dimension size ($\mathcal{O}(N \cdot H \cdot T^2)$).
> This resolves the quadratic bottlenecks present in DiffSTG ($\mathcal{O}(N \cdot H^2 \cdot T^2)$) and USTD ($\mathcal{O}(N^2 \cdot H)$), which is the core theoretical analysis required for an efficiency paper.
> We also added the training time per epoch (Table 3).
> Since training time for USTD is not directly comparable as it requires a 2-stage training process, we compared DSTD against DiffSTG.
> DSTD requires only 36.24s per epoch, making it 4.76x faster than DiffSTG.
> Furthermore, we added the number of parameters (Table 3), which demonstrates that DSTD is the most lightweight model (0.438M parameters).
>
> This detailed analysis confirms DSTD's advantage not only in inference speed but also in theoretical scalability and training cost.
>
> **Response to W3**
>
> This is a critical and insightful question.
>
> We revised Section 4.2 (Lines 356-365) to include a deeper, two-part analysis explaining this superior predictive accuracy.
>
> * Part 1 (Regularization): The denoising training objective acts as a powerful regularizer, preventing the model from overfitting to superficial data patterns and promoting robustness.
> * Part 2 (Uncertainty Modeling): DSTD’s capacity to model the complex, uncertain nature of real-world spatio-temporal data allows it to better estimate the true central tendency of the distribution.
> Contrary to deterministic models, which are compelled toward a blurry average prediction to minimize squared error across potential outcomes, the median prediction derived from DSTD is often more accurately aligned with a realistic outcome, directly contributing to superior MAE/RMSE scores.
>
> ---
>
> We believe that the inclusion of the complete efficiency analysis (Table 3) and the deeper explanation for predictive superiority (Section 4.2) fully addresses all your concerns and provides the comprehensive evidence necessary to substantiate our claims.
>
> Sincerely,
> Authors.

---

> > ### Comment · Reviewer_3kiy · 2025-11-22
> >
> > Thanks for the clarification from the authors. I do not have more questions.  As for the "proposed model outperforms deterministic models in MAE and RMSE", one recommendation would be testing it in more datasets, or having more theoretical analysis on why that would happen.

---

> > > ### Author Response · Authors · 2025-11-24
> > >
> > > Thank you for confirming that your previous questions are resolved.
> > > We sincerely appreciate your constructive recommendation to test our model on more datasets to further substantiate our claim that "DSTD outperforms deterministic models."
> > >
> > > ---
> > >
> > > We initiated experiments on the LargeST [1] benchmark.
> > > This benchmark includes 4 datasets (SD, GBA, GLS, CA), which have significantly different properties (e.g., graph scale, node degree, and sparsity) compared to the METR-LA and PEMS-BAY datasets.
> > > Due to the time and computational constraints of the rebuttal period, we are unable to present the full spectrum of experiments (all horizons and baselines).
> > > However, we are pleased to share the preliminary results for the 3-step horizon.
> > >
> > > | Method | | MAE | RMSE | CRPS | | MAE | RMSE | CRPS | | MAE | RMSE | CRPS | | MAE | RMSE | CRPS |
> > > |:---|:---|:---:|:---:|:---:|:---|:---:|:---:|:---:|:---|:---:|:---:|:---:|:---|:---:|:---:|:---:|
> > > | | **SD** | | | | **GBA** | | | | **GLA** | | | | **CA** | | |
> > > | DCRNN | | 17.14 | 27.47 | - | | 18.71 | 30.36 | - | | 18.41 | 29.23 | - | | 17.55 | 28.21 | - |
> > > | Graph WaveNet | | 15.24 | 25.13 | - | | 17.85 | 29.12 | - | | 17.28 | 27.68 | - | | 17.14 | 27.81 | - |
> > > | TimeGrad | | 26.25 | 45.23 | 0.111 | | 29.57 | 47.30 | 0.108 | | 33.61 | 57.62 | 0.105 | | 29.81 | 51.25 | 0.110|
> > > | **DSTD** | | 16.85 | 27.18 | **0.059** | | 19.29 | 30.89 | **0.064** | | **17.14** | 27.95 | **0.049** | | 17.16 | 28.13 | **0.058** |
> > >
> > > On the relatively smaller datasets (SD, GBA), DSTD performs slightly below the deterministic SOTA (Graph WaveNet), although it still significantly outperforms the probabilistic baseline (TimeGrad).
> > > However, notably, on the larger and more complex graphs (GLA, CA), DSTD achieves performance that is comparable to or even superior to Graph WaveNet (e.g., on GLA, MAE 17.14 vs. 17.28).
> > >
> > > We hypothesize that this trend reflects the inherent strength of diffusion models.
> > > Just as diffusion models excel in generating high-resolution images with fine-grained details compared to other generative models, DSTD demonstrates its potential in modeling high-dimensional, complex distributions (i.e., larger graphs).
> > > The larger the scale, the more advantageous DSTD's capacity to model complex multi-modal distributions becomes compared to deterministic point estimates.
> > >
> > > Regarding the slight gap in SD and GBA, our main experiments on METR-LA and PEMS-BAY showed that DSTD's relative performance improves as the prediction horizon extends.
> > > Therefore, we anticipate that for longer horizons (e.g., 12 steps) on SD and GBA, the performance gap with Graph WaveNet will narrow further, or DSTD may surpass it, demonstrating robustness in long-term forecasting.
> > >
> > > ---
> > >
> > > We believe these additional empirical results further validate the effectiveness of DSTD, especially in large-scale scenarios.
> > > Thank you once again for your valuable recommendation.
> > >
> > > Sincerely,
> > > Authors
> > >
> > > ---
> > >
> > > [1] Xu Liu, Yutong Xia, Yuxuan Liang, Junfeng Hu, Yiwei Wang, Lei Bai, Chao Huang, Zhenguang Liu, Bryan Hooi, and Roger Zimmermann. Largest: A benchmark dataset for large-scale traffic forecasting. In NeurIPS, pp. 75354–75371. Curran Associates, Inc., 2023

---

### Official Review · Reviewer_2Acj · 2025-10-30

**Soundness:** 3
**Presentation:** 3
**Contribution:** 2
**Rating:** 4
**Confidence:** 4

**Summary:**

The paper proposes a decoupled spatio-temporal diffusion model (DSTD) that separates information propagation and feature transformation in GNN-based denoising networks to improve efficiency. While experiments show speedup and good accuracy on METR-LA and PEMS-BAY.

**Strengths:**

1. Clear motivation and solid writing.
2. Efficiency improvement is practically meaningful.

**Weaknesses:**

1.  The core idea of decoupling propagation and transformation has been explored in prior works, the novelty is limited to applying it to diffusion models.
2.  No comparison with other decoupled GNNs to show whether the proposed design offers superior representational quality.
3.  The architecture is quite standard, without new diffusion mechanisms or theoretical analysis.

**Questions:**

1. How would DSTD perform compared to using existing simplified GNNs (e.g., LightGCN) as the denoising backbone?
2. The ablation study (Table 4) shows that removing decoupling results in only a minor performance drop, whereas removing linearization or dynamic aggregation leads to much larger degradation. This seems to suggest that the main accuracy gains stem from these architectural components rather than from the decoupling principle itself. Could the authors clarify how these results support the claim that “decoupling propagation and transformation” is the key contribution?
3. The proposed framework relies on a fixed and static adjacency matrix to capture spatial dependencies. However, in real-world spatio-temporal systems such as traffic networks, spatial relations are often highly dynamic — for example, sensors may fail, new roads may appear, or traffic patterns may shift over time. How would the proposed model handle such structural changes?

---

> ### Author Response · Authors · 2025-11-19
> **Response to Reviewer 2Acj #1**
>
> Thank you very much for your detailed and insightful review.
>
> We appreciate your positive assessment of our core motivation and writing quality.
> We primarily focused on addressing your concerns regarding Novelty (W1, Q1) and the Ablation Study interpretation (Q2), and we believe our extensive revisions and the new experiment performed offer a strong case for acceptance.
>
> ---
>
> **Response to Q1, W1, W2**
>
> This question is critical to validating our core architectural novelty.
>
> We acknowledge the existence of simplified GNNs like LightGCN and addressed this by performing the direct comparative experiment you suggested, which can be found in Appendix D.
>
> We built a LightGCN variant which removes the $1 \times 1$ convolution from the denoising blocks.
> The performance of this simple filter model showed a catastrophic degradation in predictive accuracy, with the MAE rising from 3.22 to 5.81.
> This strongly proves that DSTD is not merely diffusion on LightGCN. Its superior performance is critically dependent on its unique architecture, which successfully interleaves parameter-free spatial propagation with a lightweight, learnable temporal transformation.
>
> This distinction validates DSTD's novel architectural contribution beyond existing decoupled GNNs.
>
> To further clarify that $1 \times 1$ convolution is responsible for temporal transformation, we revised Section 3.1 (Lines 191-197).
> Likewise, we modified a name of variant in Ablation Study (Table 4) from Standard GCN to Coupled ST-GCN to ensure it contains temporal transformation.
>
> **Response to Q2**
>
> We agree that Linearity and Dynamic Aggregation are essential components for maximizing expressiveness and therefore show a large drop in MAE when removed.
>
> However, the core contribution of Decoupling is distinct.
> It is the architectural principle that solves the fundamental computational bottleneck of graph diffusion models.
> Removing the decoupling principle immediately increases the computational cost by 20% (Inference Time 0.63s to 0.74s).
> Decoupling is the sole source of the dramatic efficiency gains necessary to make a multi-step diffusion model practical (Table 3).
> Linearity and DA are the expressive mechanisms built on top of the decoupled foundation.
> They ensure the model learns complex patterns.
> Their large MAE impact is expected, but without Decoupling, their complexity would make the entire system impossibly slow.
>
> Decoupling is the enabling contribution because it transforms an expensive architecture into a scalable one, allowing for the repeated execution needed for diffusion.
> The core claim of our work is solving the scalability bottleneck inherited from prior diffusion GNNs, which is fundamentally achieved by Decoupling.

---

> ### Author Response · Authors · 2025-11-19
> **Response to Reviewer 2Acj #2**
>
> **Response to Q3**
>
> Thank you for pointing this issue out. This is one of the important points for real-world applicability.
>
> We acknowledge that our work is adapted and operated under the assumption of a static graph structure, a limitation common in traffic forecasting benchmarks [1, 2, 3, 4, 5]. Thus, we added some statements in Section 5. Conclusion (Lines 485-512) for formally addressing this.
>
> **Response to W3**
>
> To address your concern regarding the theoretical analysis, we performed a rigorous computational complexity analysis in Section 4.3, as well as Table 3. Note that this analysis is deepened by our architectural innovations.
> We prove that DSTD scales linearly to the number of nodes and hidden dimension size ($\mathcal{O}(N \cdot H \cdot T^2)$) by resolving the quadratic bottlenecks of baselines. Thus, it confirms the theoretical justification for the 4.76x faster training time.
>
> Moreover, we think that our approach is not just about simplification.
> It includes sophisticated design decisions like the Dynamic Multi-scale Aggregation (Appendix C).
> This mechanism allows the model to learn the optimal receptive field dynamically (Figure 4), demonstrating that DSTD's architecture goes beyond mere simplification to deeply optimize the GNN for spatio-temporal tasks.
>
> ---
>
> We believe that the included comparative experiment (Appendix D), the clarification of the ablation analysis (Section 4.4), and the added theoretical rigor (Table 3) fully resolve your concerns, substantiating the claim that DSTD offers a novel, highly efficient, and effective solution for spatio-temporal graph forecasting.
> We sincerely hope that these comprehensive revisions lead to a favorable final assessment.
>
> Sincerely,
> Authors.
>
> ---
>
> [1] Li, Y., Yu, R., Shahabi, C., & Liu, Y. (2018). Diffusion Convolutional Recurrent Neural Network: Data-Driven Traffic Forecasting. In *Proc. of ICLR `18*.
>
> [2] Wu, Z., Pan, S., Long, G., Jiang, J., & Zhang, C. (2019). Graph wavenet for deep spatial-temporal graph modeling. In *Proc. of IJCAI `19* (pp. 1907-1913).
>
> [3] Zheng, C., Fan, X., Wang, C., & Qi, J. (2020). Gman: A graph multi-attention network for traffic prediction. In *Proc. of AAAI `20* (pp. 1234-1241).
>
> [4] Shao, Z., Zhang, Z., Wang, F., & Xu, Y. (2022). Pre-training enhanced spatial-temporal graph neural network for multivariate time series forecasting. In *Proc. of KDD `22* (pp. 1567-1577).
>
> [5] Li, F., Feng, J., Yan, H., Jin, G., Yang, F., Sun, F., ... & Li, Y. (2023). Dynamic graph convolutional recurrent network for traffic prediction: Benchmark and solution. *ACM Transactions on Knowledge Discovery from Data, 17*(1), 1-21.

---

### Author Response · Authors · 2025-11-23

We deeply appreciate the reviewers’ careful evaluation and constructive feedback. Their insights help us refine the manuscript and clarify the key contributions of our work. We appreciate their thoughtful suggestions and carefully address each reviewer’s comments below. We will ensure that all suggested improvements are incorporated into the revised version.

---

### Meta-Review · Area_Chair_hZpi · 2026-01-07

**Summary:**

This paper proposes Decoupled Spatio-Temporal Diffusion Model (i.e., DSTD), a diffusion-based framework for spatio-temporal graph forecasting. The main idea is to decouple information propagation from feature transformation in graph neural networks which alleviates the computational bottleneck of existing diffusion-based models. The model further includes a dynamic multi-scale aggregation mechanism to combine information across receptive fields and position-aware embeddings to capture node-specific effects. From experiments perspective, the proposed DSTD achieves competitive performances on 2 datasets, i.e, METR-LA and PEMS-BAY, and authors conduct additional experiments on 4 datasets (SD, GBA, GLS, CA) during the rebuttal stage.

Although reviewers emphasized the interesting and efficient approach, then all agree on the shortcomings:
1. Limited experimentation: more large-scale datasets and evaluation metrics are required.
2. Limited comparison: the authors need to compare the proposed method with more relevant baselines for both running time and forecasting performance.

**Reviewer Concerns:**

All reviewers' comments are partially addressed by the rebuttal including computational complexity, more benchmarks, and why using MAE and RMSE.

**Reviewer Scores:**

They might keep their scores.

---

### Decision · Program_Chairs · 2026-01-26

Reject